# Bmi1 marks distinct castration-resistant luminal progenitor cells competent for prostate regeneration and tumour initiation

Young A. Yoo[1], Meejeon Roh[1,2], Anum F. Naseem[1], Barbara Lysy[1], Mohamed M. Desouki[3], Kenji Unno[1] & Sarki A. Abdulkadir[1,2,4]

Identification of defined cell populations with stem/progenitor properties is key for understanding prostate development and tumorigenesis. Here we show that the polycomb repressor protein Bmi1 marks a population of castration-resistant luminal epithelial cells enriched in the mouse proximal prostate. We employ lineage tracing to show that these castration-resistant Bmi1-expressing cells (or CARBs) are capable of tissue regeneration and self-renewal. Notably, CARBs are distinct from the previously described luminal castration-resistant Nkx3.1-expressing cells (CARNs). CARBs can serve as a prostate cancer cell-of-origin upon *Pten* deletion, yielding luminal prostate tumours. Clonal analysis using the *R26R-confetti* allele indicates preferential tumour initiation from CARBs localized to the proximal prostate. These studies identify Bmi1 as a marker for a distinct population of castration-resistant luminal epithelial cells enriched in the proximal prostate that can serve as a cell of origin for prostate cancer.

[1] Department of Urology, Northwestern University Feinberg School of Medicine, Chicago, Illinois 60611, USA. [2] The Robert H. Lurie Comprehensive Cancer Center, Northwestern University Feinberg School of Medicine, Chicago, Illinois 60611, USA. [3] Department of Pathology, Microbiology and Immunology, Vanderbilt University School of Medicine, Nashville, Tennessee 37215, USA. [4] Department of Pathology, Northwestern University Feinberg School of Medicine, Chicago, Illinois 60611, USA. Correspondence and requests for materials should be addressed to S.A.A. (email: sarki.abdulkadir@northwestern.edu).

Understanding the cellular origins of primary and castration-resistant prostate cancer (CRPC) is crucial to our efforts in improving cancer prevention and treatment. Yet in prostate cancer, our knowledge of the normal epithelial cell lineage relationships and the identities of cells that serve as targets for cancer initiation and recurrence following therapy is incomplete. The development and maintenance of both the prostate and prostate carcinoma are crucially dependent on androgens, making the prostate an excellent system to analyse stem/progenitor cell function in the context of normal development, regeneration or tumorigenesis. The adult prostate consists of three epithelial lineages: basal cells, identified by cytokeratins CK5, CK14 and p63; secretory luminal cells expressing CK8, CK18 and androgen receptor; and rare neuroendocrine cells expressing synaptophysin and chromogranin A[1]. Previous studies have indicated that stem/progenitor cells exist in both the basal and luminal cell compartments of the prostate[2–5]. Lineage tracing and tissue recombination studies have shown that basal cells in the adult prostate exhibit bipotentiality and self-renewal capacity during regeneration and tissue homeostasis[6–10]. During prostate postnatal development, basal cells undergo asymmetric division and generate one stem cell and one progenitor cell that differentiates to a luminal cell[11,12]. By contrast, a number of lineage-tracing studies have shown that basal and luminal cell lineages in the adult murine prostate are mostly self-sustained[10,13].

Although prostate adenocarcinoma displays a strong luminal phenotype, both prostate basal and luminal cells can serve as cells of origin for prostate cancer, although basal cells may first differentiate into luminal cells before transformation[5,10,13–15], highlighting the difference between a cell of mutation and a cell of origin for cancer. Furthermore, evidence from multiple mouse models suggests that luminal cells are favored as a cell-of-origin for prostate cancer[16,17]. In adult mouse prostate, Shen and colleagues[5] identified a rare luminal population of castration-resistant Nkx3.1-expressing cells (CARNs) that displays stem cell properties and serves as an efficient cell of origin for prostate cancer in vivo.

Bmi1, which was identified as a target of Molony virus insertion in B-lymphoid tumours of Eμ-Myc transgenic mice, is a member of polycomb-repressing complex 1 that has an essential role in maintaining chromatin silencing[18]. Bmi1 has a key role in self-renewal and tumorigenesis in various tissues, through mechanisms that are both dependent and independent of repression of the cell cycle regulators p16[Ink4a] and p19[ARF] (refs 19,20). In human prostate cancer, BMI1 is often upregulated and associated with tumour progression and poor prognosis[21–23]. Lukacs et al.[24] recently used a tissue recombination system with dissociated prostate epithelial cells to show that Bmi1 expression is important for prostate regeneration and progression of Pten loss-initiated cancer. However, whether Bmi1 marks cells that are competent for prostate regeneration and tumour initiation in intact tissues in vivo has not been examined. In this study, we employed lineage tracing to show that Bmi1-expressing cells mark a distinct, largely luminal castration-resistant prostate epithelial cell population that is capable of prostate regeneration and cancer initiation.

## Results

**Bmi1 expression in luminal cells of the proximal prostate.** We first examined the expression pattern of Bmi1 protein in mouse prostate tissues by immunohistochemistry, using the known pattern of Bmi1 expression in the intestinal epithelium as a positive control (Supplementary Fig. 1a). In the adult prostate, we divided the prostate gland into proximal, intermediate and distal

thirds and found that most Bmi1-expressing cells localized to the proximal region of the gland (Supplementary Fig. 1b–g). Notably, a higher percentage of CK8-expressing luminal cells coexpressed Bmi1 compared with cells expressing the basal cell marker p63. In the anterior prostate, 60% of CK8 + cells and 21.6% of p63 + cells coexpressed Bmi1 (Supplementary Table 1). Additionally, more Bmi1 + cells in the intact anterior prostate coexpressed CK8 versus p63 (93% versus 7.5%), CK14 (97.5% versus 2.5%) or CK5 (97.9% versus 2.1%) (Supplementary Table 1). In the regressed anterior prostate following castration, 1.9% of epithelial cells expressed Bmi1 with a majority coexpressing the luminal marker CK8 compared with the basal markers CK14 (98% versus 2%), CK5 (97.6% versus 2.4%) or p63 (93.3% versus 8.3%) (Supplementary Fig. 1h,i and Supplementary Table 1). As an earlier report had suggested that Bmi1 + cells are mainly localized to the Sca-1 + basal cell compartment of the proximal mouse prostate[24], we examined this issue further using Bmi1-GFP (green fluorescent protein) knock-in mice that express GFP under control of the endogenous Bmi1 regulatory region[25]. We found expression of GFP in both luminal and basal cell fractions by immunohistochemistry and by flow cytometry (Supplementary Fig. 2). Flow sorting of Lin⁻Sca1⁻CD49f[lo] luminal and Lin⁻Sca1⁺CD49f[hi] basal cells revealed that GFP + cells represented 31.7% of the Lin⁻Sca1⁻CD49f[lo] luminal cell fraction and 10.4% of the Lin⁻Sca1⁺CD49f[hi] basal cell fraction, in a 3:1 ratio, which is similar to that of Bmi1 + luminal to Bmi1 + basal cells by immunohistochemistry (Supplementary Fig. 2d–f and Supplementary Table 1). We also examined Bmi1 expression in sorted luminal and basal cells from wild-type mice (Supplementary Fig. 2g). Lin⁻Sca1⁻CD49f[lo] luminal cells expressing high levels of Krt8 and low level of Tp63 robustly expressed Bmi1 compared with Lin⁻Sca1⁺CD49f[hi] basal cells (Supplementary Fig. 2h). To determine if Bmi1-expressing cells in the regressed prostate overlap with the recently described CARNs, we co-stained prostate sections with Bmi1 and Nkx3.1 antibodies. We found no overlap between Bmi1 + cells and CARNs (n = 807 Bmi1 + cells counted from a total of 41,813 cells from five mice; Supplementary Fig. 1j). Thus luminal castration-resistant Bmi1-expressing cells (or CARBs) appear to be distinct from CARNs.

**Lineage marking of Bmi1-expressing cells in the prostate.** To trace the fate of Bmi1 + cells in the prostate, we made use of the Bmi1-Cre[ER] allele in which the inducible Cre[ER] fusion protein gene was 'knocked' into the 3′ untranslated region of the mouse Bmi1 gene[26], in conjunction with the R26R-YFP (yellow fluorescent protein) reporter[27]. Notably, the Bmi1-Cre[ER] knock-in allele does not impair endogenous Bmi1 expression[26], We induced Cre[ER] activity by tamoxifen treatment of hormonally intact adult male Bmi1-Cre[ER];R26R-YFP (hereafter BY) mice for 4 days, then examined tissues 3 weeks later (Fig. 1a). As a control, we examined the duodenum and pancreas, which are known to contain Bmi1 + stem cells, and identified clusters of YFP-marked cells in both organs (Fig. 2b,c). Examination of the prostates of tamoxifen-treated mice revealed YFP-marked cells scattered throughout the gland at similar frequencies in the anterior (0.3%), ventral (0.3%), and dorsolateral (0.4%) prostate lobes (n = 90,888 total cells counted from five mice; Supplementary Table 2). Furthermore, we observed an enrichment of YFP-marked cells in the proximal ducts adjacent to the urethra compared with the intermediate and distal regions (Fig. 2d,e and Supplementary Table 2), consistent with our Bmi1 immunohistochemical staining data and analysis of Bmi1-GFP knock-in mice. The majority (95%) of YFP-marked cells were luminal, expressing CK8 or androgen receptor, while

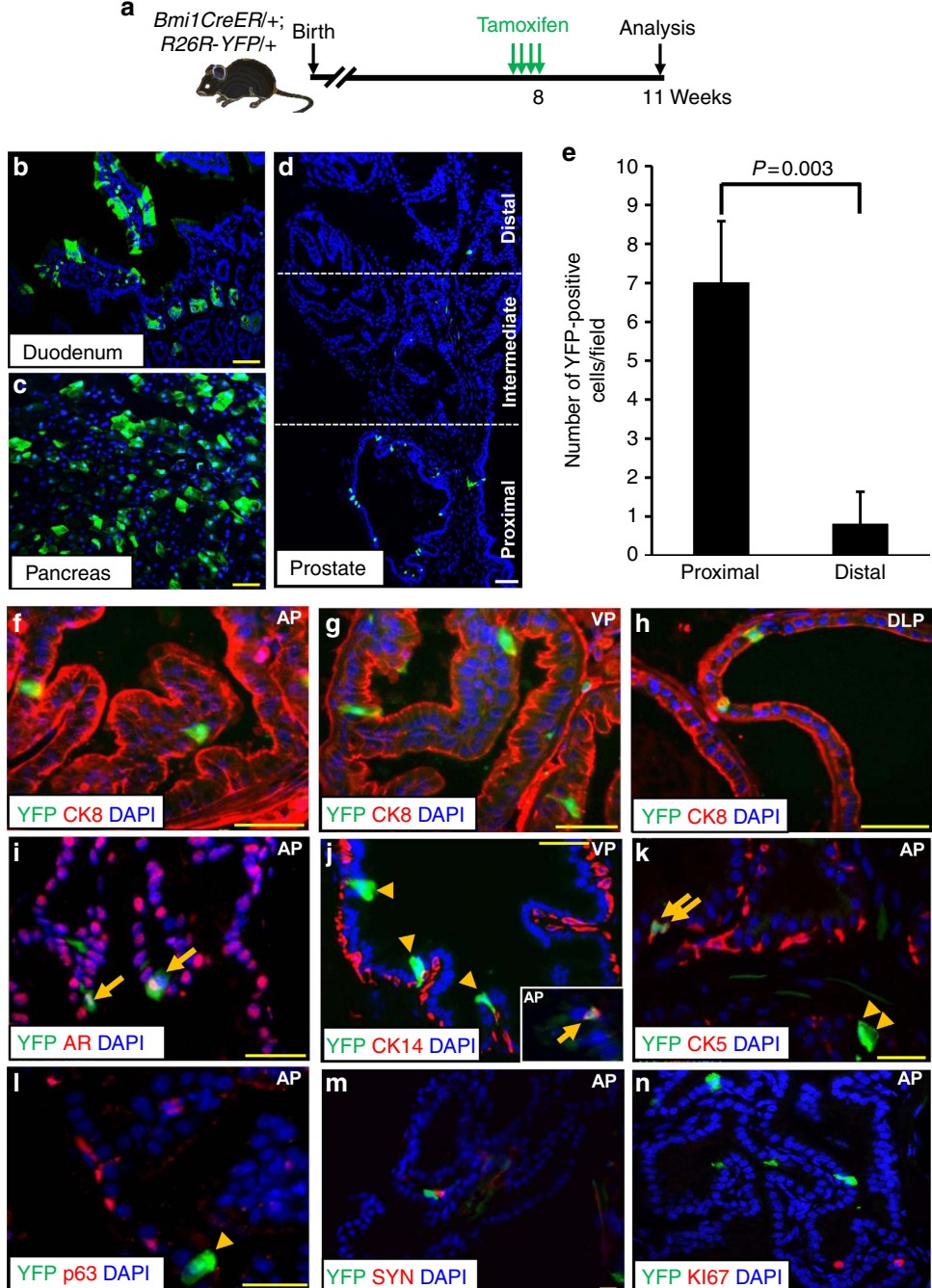

**Figure 1 | Bmi1 is largely expressed in the luminal cells of the proximal mouse prostate.** (**a**) Scheme for lineage marking in hormonally intact mice. (**b–d**) YFP expression (green) in duodenum (**b**), pancreas (**c**) and prostates (**d**) after 3 weeks of tamoxifen treatment. Blue, DAPI. (**e**) Quantitation of YFP + cells in proximal and distal prostates of anterior prostate ($n = 4$). Data represent the mean ± s.d., two-tailed Student's $t$-test. (**f–i**) Immunofluorescence (IF) staining shows co-localization of YFP with CK8 (**f–h**) or androgen receptor (AR) (arrows, **i**) in luminal cells. (**j–l**) Co-localization of YFP with CK14 (**j**), CK5 (**k**) or p63 (**l**) expressing basal cells. Arrows: YFP + cells coexpressing basal cell markers. Arrowheads: YFP + cells negative for basal cell markers. (**m,n**) IF shows that YFP does not co-localize with synaptophysin (SYN, (**m**)) and Ki67 (**n**), respectively. Yellow scale bar, 50 um, White scale bar,100 μm.

3–4.5% of YFP + cells expressed the basal cell markers CK5, CK14, or p63 (Fig. 1f–l; $n = 199$ YFP + cells counted from three mice; Supplementary Table 2). YFP-marked cells did not express the neuroendocrine marker synaptophysin (Fig. 1m). In the anterior prostate, none of the YFP-marked cells examined coexpressed the proliferation marker Ki67 ($n = 0$ out of 199 YFP + cells counted from three mice; Fig. 1n; Supplementary Table 2), which is significantly lower than the Ki67 index in YFP-negative cells ($n = 29$ out of 6,629 YFP- cells counted from three mice; $P < 0.01$, two-tailed Student's $t$-test).

**CARBs are distinct from CARNs.** In the regressed prostate following castration, tamoxifen induction of *BY* mice resulted in the labelling of isolated luminal and basal cells, which were enriched in the proximal region (Fig. 2a–f and Supplementary Fig. 3a–c). We term these cells CARBs. CARBs represented 0.7% of the total epithelial cells in the regressed prostates ($n = 306$ YFP + cells out of a total of 40,662 cells counted from three mice; Supplementary Table 2). The majority of CARBs coexpressed the luminal markers CK8 (95.8%) or androgen receptor (95%), with only 3–4.2% coexpressing CK14, CK5 or p63 (Supplementary

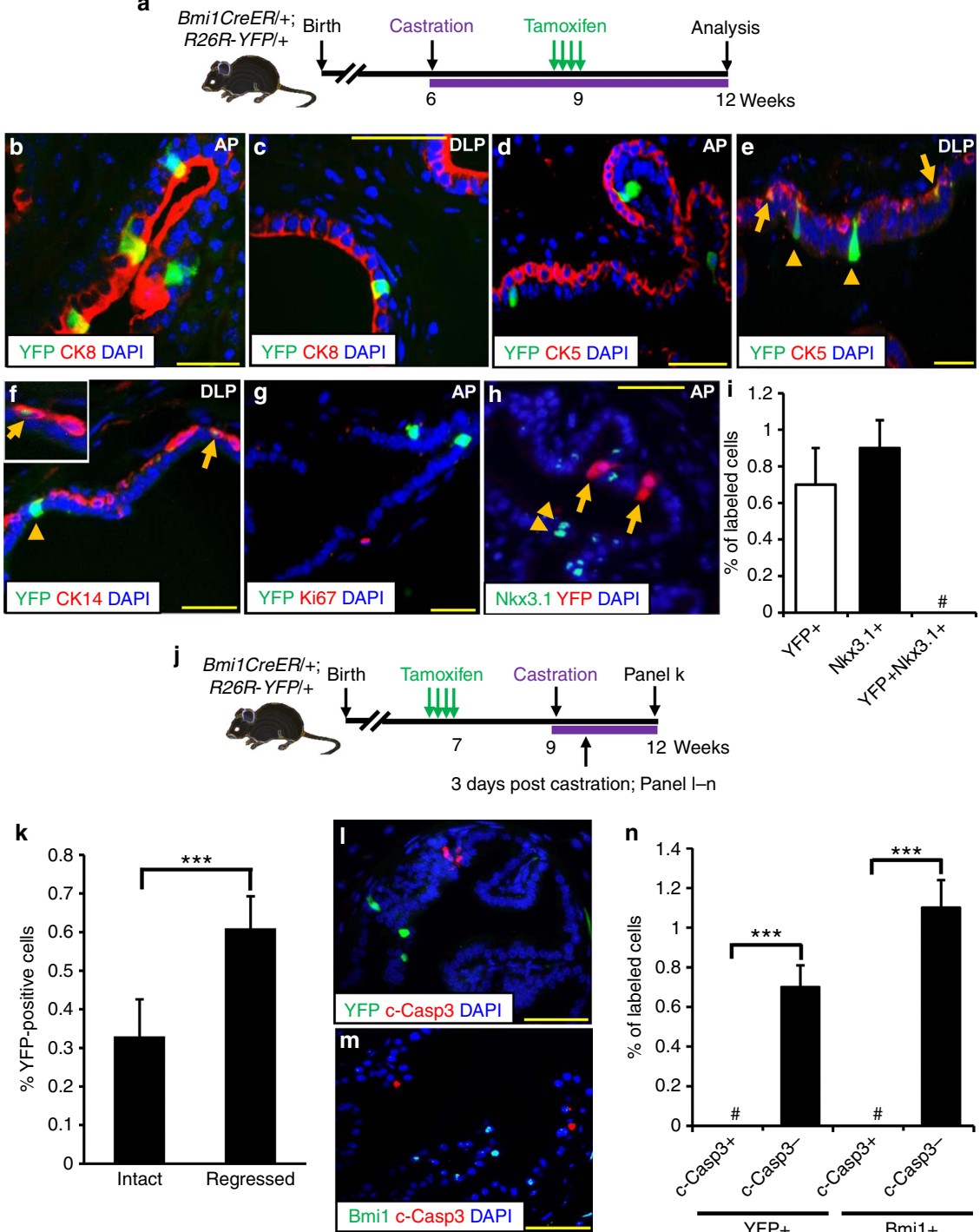

**Figure 2 | Bmi1 marks castration-resistant cells that are distinct from CARNs.** (**a**) Scheme for labelling of castration-resistant Bmi1+ cells (CARBs) in mouse prostate. (**b–f**) Co-localization of YFP in CARBs with CK8 (**b,c**), CK5 (**d,e**), or CK14 (**f**) expressing cells in regressed prostates. Arrows and arrowheads indicate YFP-labelled cells that are positive and negative for basal cell markers, respectively. (**g**) Ki67 does not co-localize with YFP+ CARBs in the regressed prostate. (**h**) YFP+ CARBs (arrows) are distinct from castration-resistant Nkx3.1-expressing cells (CARNs, arrowheads). (**i**) Bar graph shows the quantification of YFP+, Nkx3.1+, and YFP+Nkx3.1+ cells in the anterior prostate ($n = 3$). # No overlap between YFP+ cells and Nkx3.1+ cells (CARNs) was observed. (**j**) Scheme for labelling Bmi1+ cells before castration. (**k**) Graph showing percentage of total YFP+ cells in the anterior prostate of intact and castrated mice ($n = 4$). (**l,m**) No YFP+ (**l**) or Bmi1+ (**m**) cells examined coexpressed with cleaved caspase-3 (c-Casp3) in the regressed prostate. (**n**) Bar graph shows the quantification of YFP+ or Bmi1+ cells undergoing apoptosis. # No overlap between YFP+ or Bmi1+ cells and c-Casp3+ cells was observed. Data represent the mean ± s.d., $P^{***} < 0.001$, two-tailed Student's $t$-test. AP, anterior prostate; VP, ventral prostate; DLP, dorsolateral prostate; Scale bar, 50 μm.

Table 2). Notably, none of the YFP+ cells analysed in the regressed prostates coexpressed Nkx3.1, further supporting the distinct identity of CARBs from CARNS ($n = 322$ YFP+ cells and 1,682 Nkx3.1+ cells counted from five mice; Fig. 2h,i and Supplementary Table 2). Additionally, analysis of the regressed prostate from $Nkx3.1Cre^{ERT2};R26R$-YFP mouse stained for YFP

and Bmi1 proteins by immunohistochemistry showed no overlap between CARNs (YFP+) and CARBs (Bmi1+) (Supplementary Fig. 3d–h).

To directly examine whether YFP+ cells identified in intact mice by treating *BY* mice with tamoxifen are castration resistant, we first treated *BY* mice by tamoxifen treatment to label the cells then castrated the mice (Fig. 2j). After castration, the fraction of YFP+ cells increased from 0.3 to 0.64% ($n = 253$ YFP+ cells counted from a total of 39,281 cells from four mice; $P < 0.001$, two-tailed Student's *t*-test; Fig. 2k). These results suggest that tamoxifen treatment of *BY* mice before castration labels castration-resistant cells. We then examined whether the YFP+ cells are resistant to apoptosis after castration. As it had been reported that the peak of epithelial apoptosis in the mouse prostate occurs 3–4 days after castration[28,29], we evaluated apoptosis by cleaved caspase-3 immunodetection 3 days after castration of tamoxifen treated *BY* mice (Fig. 2j). The overall percentage of apoptotic cells by 3 days of castration was 4.8% ($n = 387$ c-casp3+ cells counted from a total of 7,994 cells from four mice). Notably, no YFP+ or Bmi1+ cells coexpressing cleaved caspase-3 were observed ($n = 242$ YFP+ and 862

Bmi1+ cells counted from four mice, respectively; Fig. 2l–n), demonstrating the relative castration resistance of YFP+ cells prospectively marked in intact adult animals. Altogether, our data point to the existence of a distinct population of castration-resistant cells or CARBs that are predominantly luminal and marked by Bmi1 expression and lack of Nkx3.1 expression.

**CARBs contribute to prostate regeneration and can self-renew.** To trace the fate of CARBs during regeneration of the adult prostate, we used the well-established prostate regression/regeneration paradigm. We castrated *BY* mice, labelled CARBs by tamoxifen treatment, and then induced prostate regeneration by treating mice with dihydrotestosterone (DHT). The regression/regeneration process was repeated for two or three rounds (Fig. 3a). The fraction of YFP+ cells increased fourfold from 0.6 to 2.8% with one round of prostate regeneration (Fig. 3b–f, Supplementary Table 3), indicating the potential of CARBs to contribute to prostate regeneration. Both luminal and basal cells were labelled with YFP in castrated mice and the relative ratios of luminal to basal YFP-labelled cells remained constant in intact,

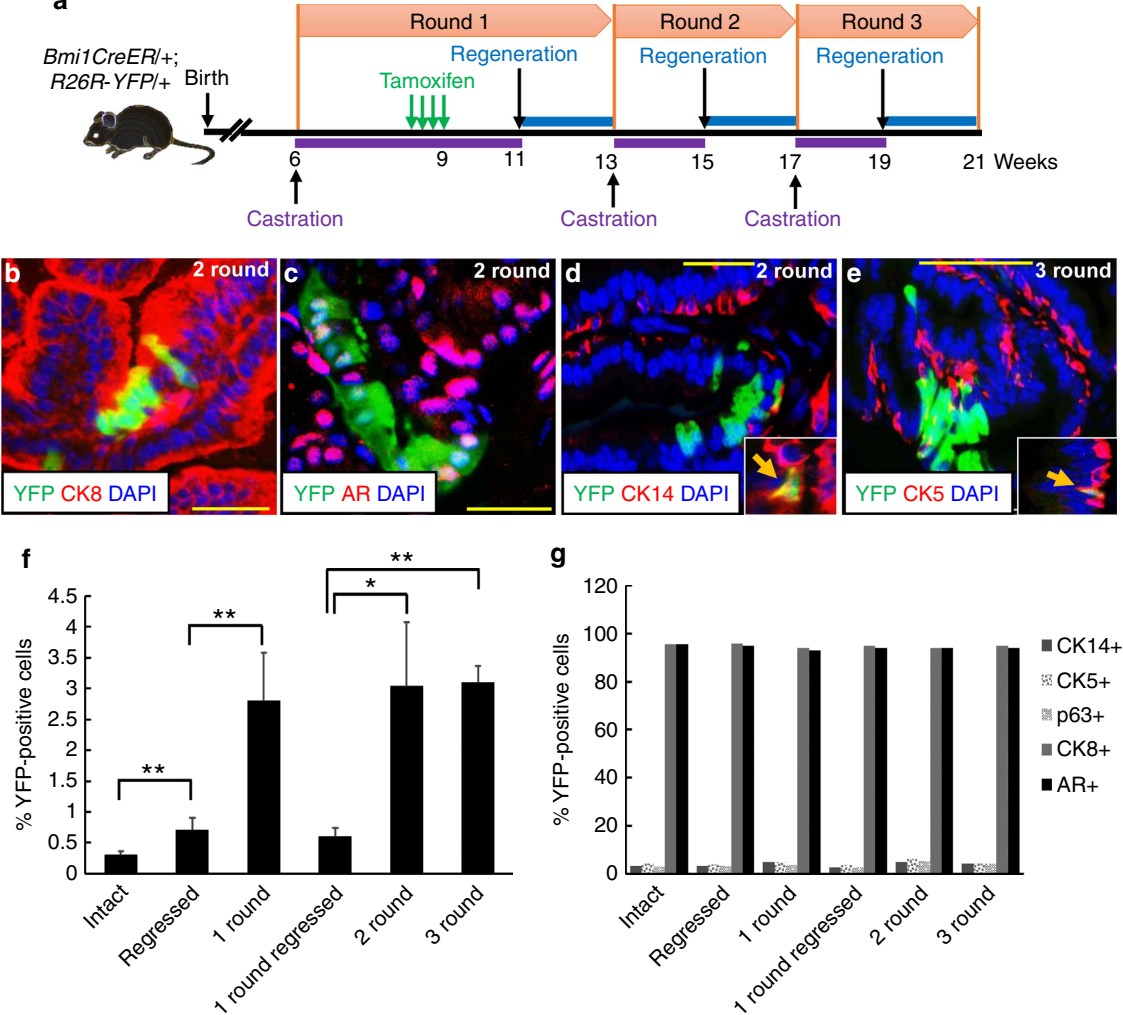

**Figure 3 | CARBs have regenerative capacity *in vivo*.** (**a**) Scheme for lineage marking during serial prostate regression and regeneration. (**b–e**) Immunofluorescence staining to assess YFP+ luminal CARBs that coexpress CK8 (**b**) and androgen receptor (AR) (**c**) or YFP+ basal CARBs that coexpress CK14 (arrow, **d**) or CK5 (arrow, **e**) after two (**b–d**) or three (**e**) rounds of serial prostate regeneration. (**f**) Graph showing percentage of total YFP+ cells in intact, castrated or regenerated prostates in the anterior prostates. Data represent the mean ± s.d. *$P < 0.05$, **$P < 0.01$, two-tailed Student's *t*-test. (**g**) Graph showing percentage of YFP+ cells coexpressing luminal or basal markers in intact, castrated or regenerated prostates. Scale bar, 50 μm.

castrated and regenerated prostates (Fig. 3g, Supplementary Table 3), suggesting that the labelled Bmi1 + basal and luminal cells (CARBs) are self-sustaining and can support long-term self-renewal. As the overall fraction of YFP + cells did not increase further with additional rounds of regression/regeneration (Fig. 3f), we concluded that while CARBs can regenerate prostate epithelial cells following castration, some of the new progeny they produce are sensitive to castration. Indeed, analysis of re-castrated mice after one round of regression/regeneration confirmed this as the fraction of YFP + cells returned to similar levels as that in the initially labelled CARBs after the first castration (0.7% versus 0.6%, respectively, Supplementary Table 3, bar labelled '1 round regressed' in Fig. 3f). These results, together with the fourfold expansion in the number of YFP + cells after each round of prostate regeneration are consistent with a model in which CARBs undergo two cell divisions during regeneration to yield four daughters, one of which retains the CARB identity of castration-resistance. To directly demonstrate self-renewal, we castrated *BY* mice, induced regeneration while labelling dividing CARBs with BrdU (Fig. 4a). After full prostate regeneration, mice were re-castrated and CARBs that have undergone a cell division and retained a castration-resistant Bmi1-expressing phenotype were identified by their coexpression of YFP, BrdU and Bmi1 (Fig. 4b–e). These results provide evidence of self-renewal by CARBs *in vivo* to generate new CARBs.

We next investigated the kinetics of cell cycle entry and exit by basal and luminal CARBs and their progeny as compared

with the rest of prostate luminal and basal epithelial cells by strategic BrdU labelling. We castrated *BY* mice then treated them with tamoxifen 2 weeks later to mark CARBs with YFP. After an additional 2 weeks, we induced prostate regeneration by DHT treatment (Fig. 5a). Analysis of BrdU incorporation at 3 days post-DHT provides insight into cell cycle entry while BrdU incorporation at 7 days post-DHT monitors exit from the cell cycle and terminal differentiation prostate epithelial cells[30]. Overall, luminal cells had a higher BrdU-incorporating rate than basal cells at both 3 days and 7 days following DHT administration, indicating increased proliferation by luminal (likely transit amplifying) cells during regeneration (Fig. 5b–d; Supplementary Table 3). Next, we examined cell cycle entry and exit in YFP-labelled cells. We analysed only YFP-labelled luminal cells as the number of YFP + basal cells were too few to allow meaningful analysis. Notably, luminal YFP + cells incorporated BrdU at 3 days post-DHT treatment (cell cycle entry) at a rate similar to all luminal cells (Fig. 5d and Supplementary Table 3). However, luminal YFP + cells had a significantly lower level of BrdU incorporation at 7 days post-DHT (1.2% versus 2.5%, $P < 0.05$, two-tailed Student's *t*-test) (Fig. 5d and Supplementary Table 3), which may be an indication that the YFP-labelled luminal cell population contains proportionately less transit amplifying cells. Furthermore, the ratio of dividing YFP + luminal to dividing YFP + basal cells was similar to the ratio of total YFP + luminal to total YFP + basal cells at 3 days of regeneration (Fig. 5e

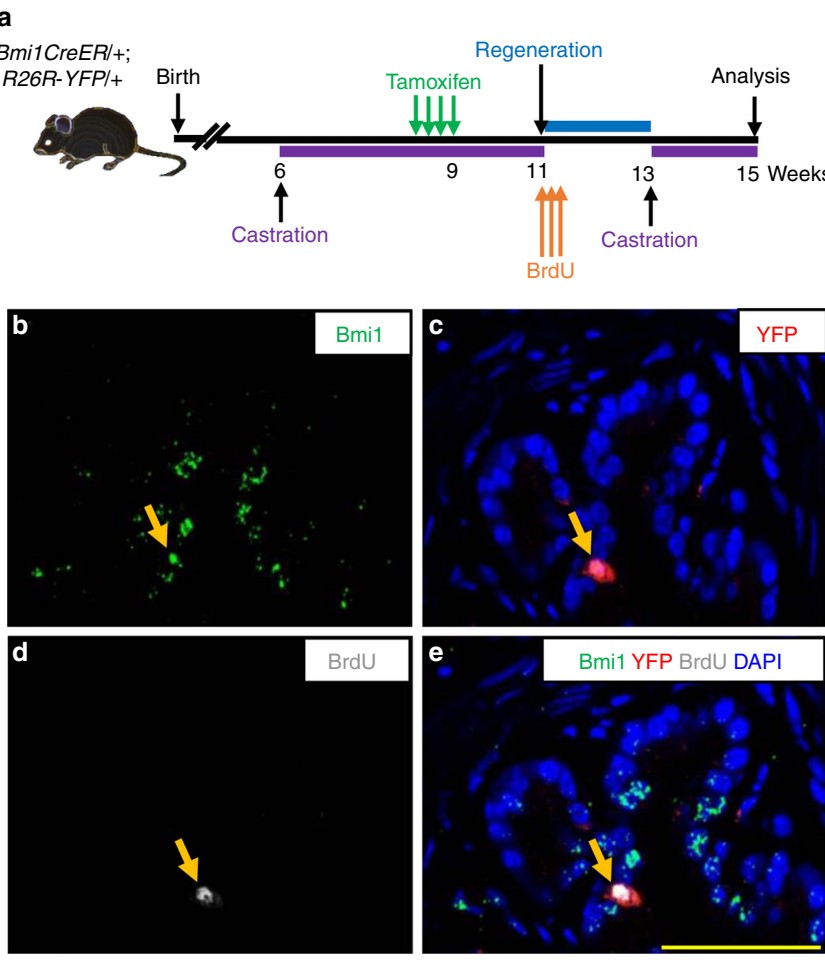

**Figure 4 | CARBs self-renew in vivo.** (**a**) Scheme for assessing self-renewal potential of CARBs. (**b–e**) Representative images for CARB labelled with YFP at 9 weeks, underwent cell division as determined by incorporation of BrdU at 11 weeks and retained castration-resistance identity in second regression at 13 weeks, consistent with self-renewal (images show Bmi1 + YFP + BrdU + cell analysed at 15 weeks, arrows). Scale bar, 50 µm.

and Supplementary Table 3). Although the cell numbers are low due to low recombination efficiency of the Bmi1CreER driver, these observations indicate that luminal and basal CARBs self-renew and that luminal and basal Bmi1+ lineages are self-sustained.

**CARBs are a prostate cancer cell-of-origin.** To assess the ability of Bmi1+ cells to serve as prostate cancer initiating cells, we specifically deleted *Pten* in Bmi1-expressing cells by treating *Bmi1-Cre^{ER};R26R-confetti;Ptenf/f* mice (hereafter *BC-Pten* mice) with tamoxifen (Fig. 6a). The *R26R-confetti* allele allows for the

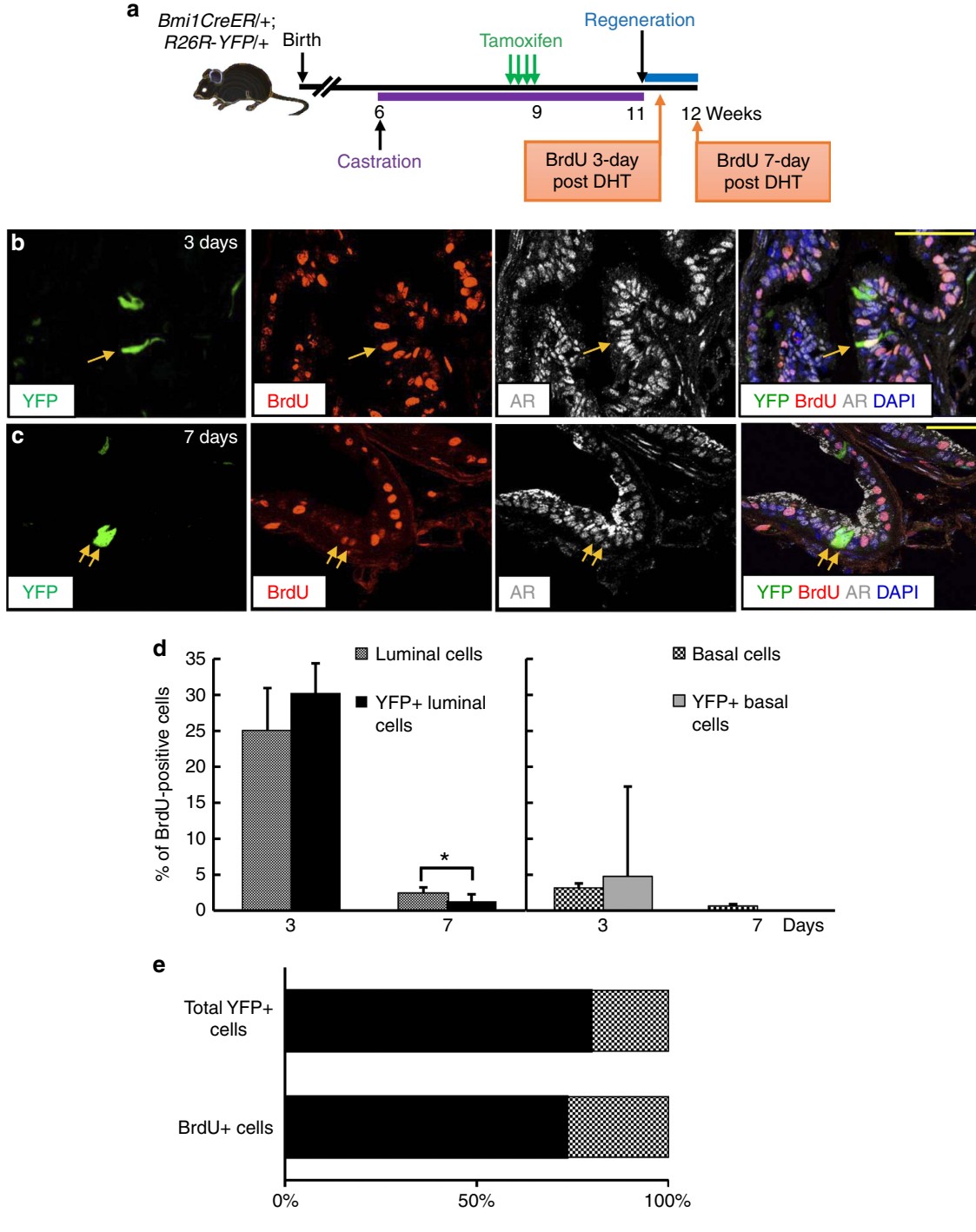

**Figure 5 | Cell cycle entry and exit of luminal and basal CARBs in regenerating prostates.** (**a**) Scheme for BrdU labelling during prostate regeneration for analysis of cell cycle entry (3 day post-DHT) and exit (7 day post-DHT). (**b,c**) Immunoflorescence for BrdU, YFP+ and luminal androgen receptor (AR) markers cells at 3 (**b**) or 7 (**c**) days after DHT-induced prostate regeneration. Arrows indicate BrdU-labelled YFP+ luminal cells. Scale bar, 50 μm. (**d**) Graphs showing percentage of all luminal (AR+) or YFP+ luminal (YFP+AR+) and all basal (p63+) or YFP+ basal (YFP+p63+) cells incorporating BrdU at 3 (*n*=3) or 7 (*n*=3) days after DHT-induced prostate regeneration. Data represent the mean ± s.d. *P<0.05, two-tailed Student's *t*-test. (**e**) Graph shows that among YFP+ cells, a similar ratio of luminal-to-basal cells enter the cell cycle at 3 days as the ratio of total luminal-to-basal cells, suggesting that YFP+ luminal and basal cells are self-sustained.

stochastic marking of mutant cells with one of four colours: nuclear GFP, cytoplasmic red fluorescent protein (RFP), cytoplasmic YFP or membrane cyan fluorescent protein (CFP) (Supplementary Fig. 4a). In our cohort, recombination was biased towards GFP (36%) and YFP (34%) relative to RFP (17%) and

CFP (13%) (Supplementary Fig. 4b). We analysed mice at 2 months and 4 months after induction of recombination (Fig. 6b–r). At 2 months, we observed focal epithelial hyperplasia and high-grade prostatic intraepithelial neoplasia (HGPIN) in the proximal region (Fig. 6h). These lesions expressed activated p-Akt

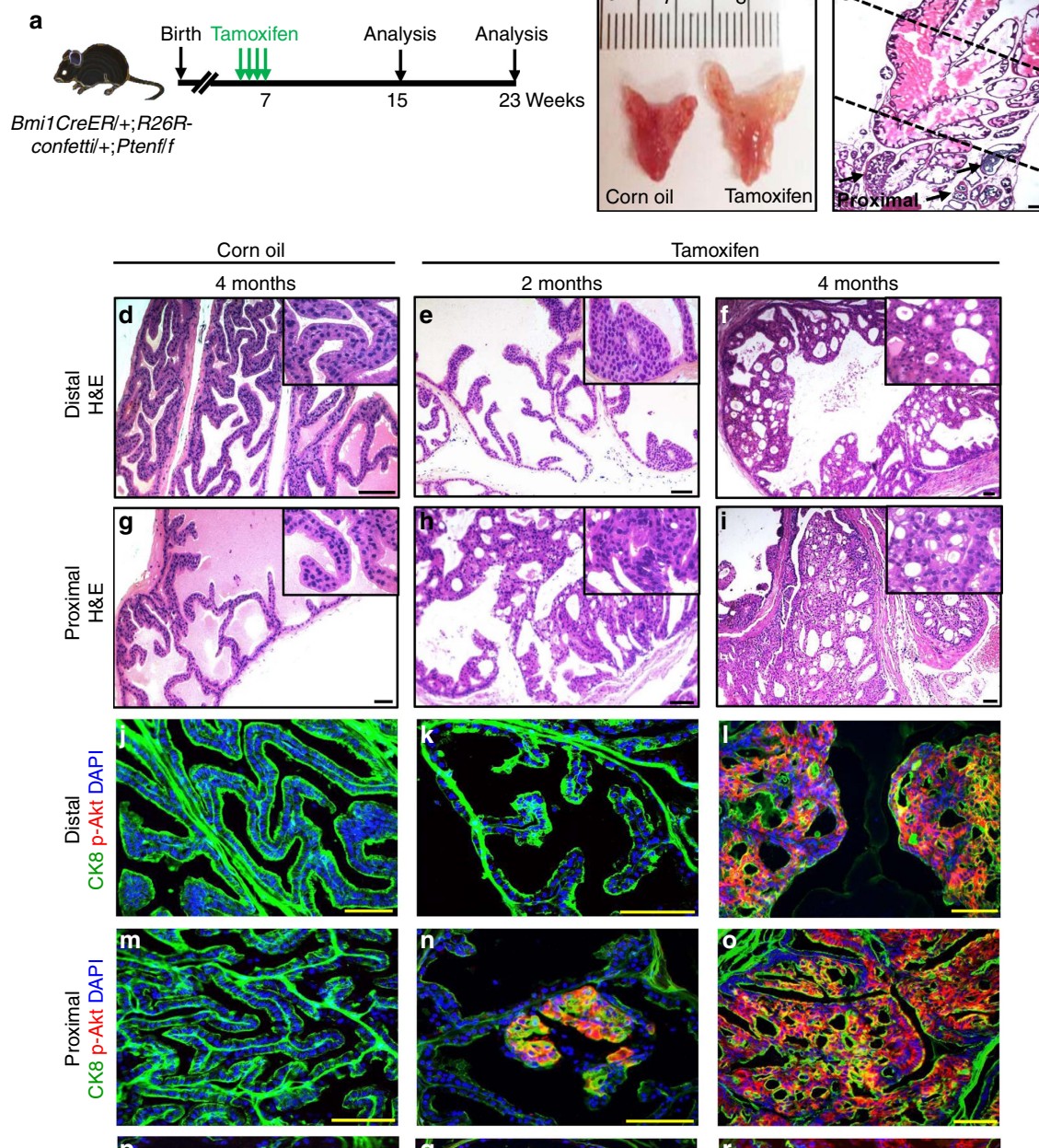

**Figure 6 | Bmi1+ cells serve as prostate cancer initiating cells.** (**a**) Scheme for prostate cancer initiation in hormonally intact *Bmi1-Cre^ER;R26R-confetti;Ptenf/f* (*BC-Pten*) mice. (**b**) Representative images of prostate lobes from *BC-Pten* mice 4 months after treatment with tamoxifen or corn oil vehicle. (**c**) H&E image showing the formation of focal epithelial hyperplasia and prostatic intraepithelial neoplasia (PIN) lesions (arrows) in the proximal prostate at 2 months post tamoxifen induction. Scale bar, 100 um. (**d–i**) H&E staining of distal (**d–f**) and proximal anterior prostate (**g–i**) at 2 (**e,h**) or 4 months (**d,g,f,i**) after tamoxifen or vehicle treatment. (**j–r**) Immunoflorescence staining for CK8 (**j–o**) or CK14 (**p–r**) with p-Akt in distal (**j–l**) and proximal prostate (**m–r**) at 2 or 4 months post tamoxifen induction. Scale bar, 50 μm.

(4.7%; $n = 152$ p-Akt + cells counted from a total of 3,251 cells) and CK8 (Fig. 6n,q). By contrast, the distal regions showed little abnormality and no expression of p-Akt (0%; $n = 0$ p-Akt + cells counted from a total of 3,788 cells) (Fig. 6e,k). By 4 months after tamoxifen induction, both distal and proximal regions showed HGPIN/cancer with a luminal phenotype and p-Akt expression (10.5%; $n = 658$ p-Akt + cells counted from a total of 6,275 cells versus 91.9%; $n = 6,455$ p-Akt + cells counted from a total of 7,024 cells) in all lobes of the prostate (Fig. 6f,i,I,o,r). Lesions were distinctly labelled with one of the four confetti colours (green, red, yellow or blue) with minimal mixing, suggesting their clonality (Fig. 7b,c; Supplementary Fig. 4h,i). Although Pten was deleted in some Bmi1-expressing basal cells, the resulting p-Akt + basal cells were detected specifically only in normal looking glandular structures (Supplementary Fig. 4j), suggesting Pten deletion in Bmi1 + basal cells does not efficiently initiate transformation. Nonetheless, we cannot completely ruled out alternative possibilities, such as basal-to-luminal differentiation as an essential step for tumour initiation.

The increased incidence of HGPIN/cancer observed in the proximal prostate at 2 months post-tamoxifen treatment suggests that either proximal Bmi1 + cells are more susceptible to *Pten*-loss induced transformation, or may simply reflect the higher density of Bmi1 + cells in the proximal region. To investigate this, we performed clonal analysis in the proximal and distal prostate regions of induced *BC-Pten* mice at 2 months and 4 months (Fig. 7). We found that at 2 months, the distal region consists of mainly singly labelled cells while the proximal region has significantly increased number of large clones consistent with expansion of mutant cells. At 4 months, the size of clones in the proximal region is further increased

significantly, while distal clones show little difference in size to clones at 2 months (Fig. 7d). We only observed singly labelled confetti cells in both distal and proximal prostates of *Bmi1Cre^{ER};R26R-confetti* intact mice early after tamoxifen induction, making it unlikely that the differences in clone size observed are due to existing coherent clones in the proximal prostate (Supplementary Fig. 4c–f). These results indicate that proximal Bmi1 + cells are more efficient at initiating prostate cancer due to *Pten* deletion.

To formally test the ability of CARBs post-castration to also serve as a cell of origin for cancer, we treated castrated *BC-Pten* mice with tamoxifen, with or without subsequent DHT treatment (Fig. 8a). The mice developed HGPIN/carcinoma with a luminal phenotype and upregulation of p-Akt (Fig. 8b–e).

As Bmi1 is not exclusively expressed in the prostate epithelium, it is possible that *Pten* deletion in non-prostate Bmi1-expressing cells, for example, in the microenvironment, might have influenced prostate cancer development in *BC-Pten* mice. To exclude this possibility, we used a tissue recombination strategy to rescue transgenic prostates in host mice. We recombined prostate epithelium from adult *BC-Pten* mice with rat urogenital mesenchyme in collagen and grafted under the renal capsule of severe combined immunodeficient (SCID) mice (Fig. 9a). Six weeks later, host animals were treated with tamoxifen to activate CreER expressed only in the Bmi1 + cells of the regenerated prostate epithelium. Grafts were then analysed at various time points after tamoxifen induction. One month post-tamoxifen treatment, grafts displayed hyperplastic foci with p-Akt expression, and by 4 months, grafts showed progressive increase in p-Akt positive lesions, with multifocal HGPIN and cancer

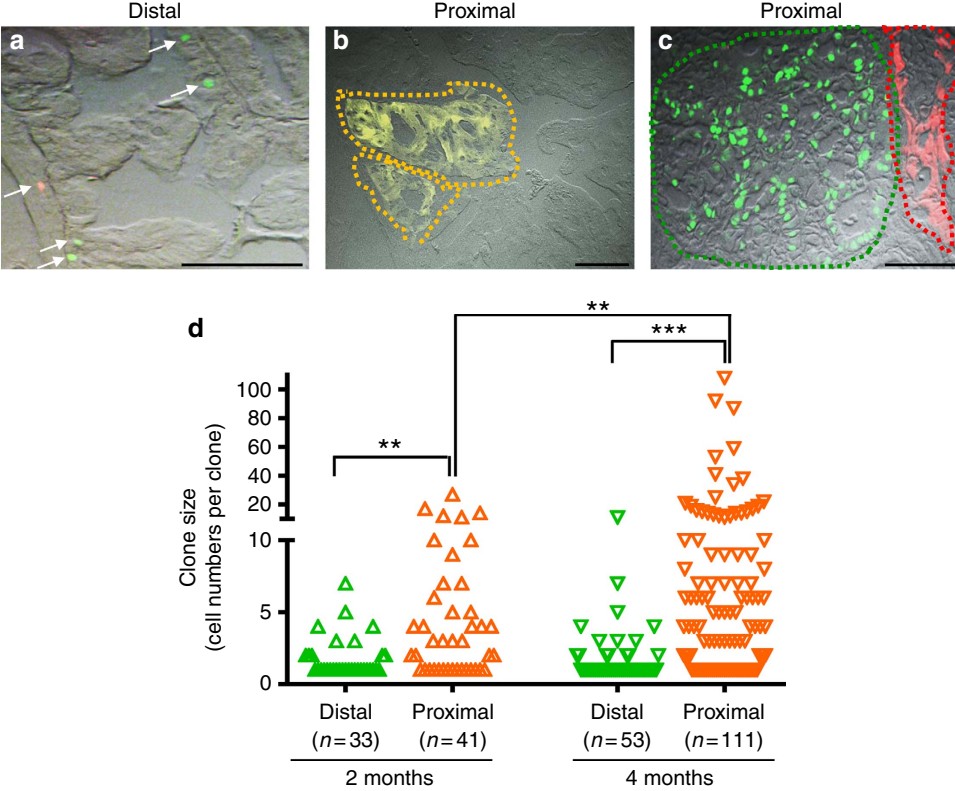

**Figure 7 | Proximal Bmi1 + cells preferentially undergo clonal expansion during prostate tumorigenesis.** (**a–c**) The distribution of distinctly labelled *confetti* clones with one of the four confetti colours (green, red, yellow or blue) in the distal (**a**) and proximal prostate (**b,c**) of *BC-Pten* mice 4 months after tamoxifen induction. Scale bar, 50 μm. (**d**) Dot plot showing change in clone size in the distal and proximal prostates over time. Clone size is defined by total number of constituent cells. The number of clones counted in each group is shown from six mice. **$P < 0.01$, ***$P < 0.001$, two-tailed Student's *t*-test.

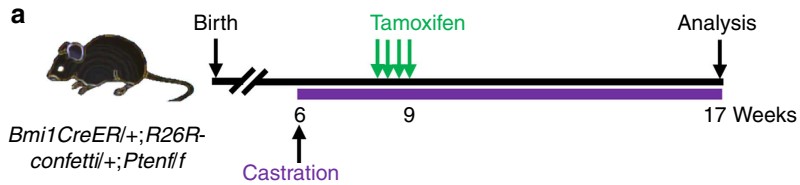

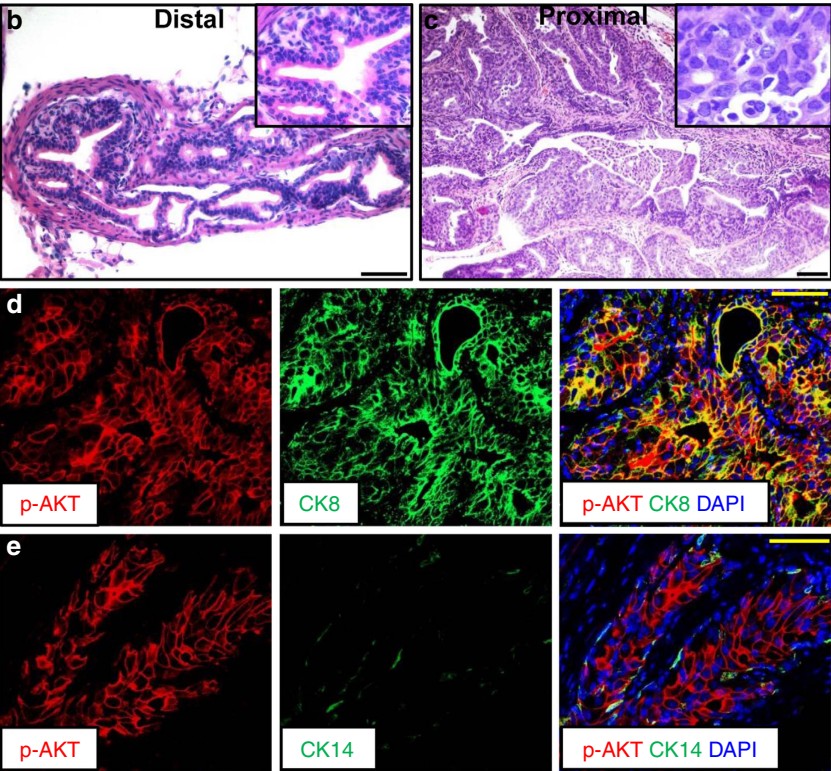

**Figure 8 | CARBs prost-castration serve as a prostate cancer cell-of-origin.** (**a**) Scheme for CARB-specific *Pten* deletion to initiate prostate cancer. (**b,c**) H&E staining of distal (**b**) and proximal prostate (**c**) at 3 months post tamoxifen induction. (**d,e**) Immunoflorescence staining for CK8 (**d**) or CK14 (**e**) with p-Akt in CARB-initiated neoplastic lesions upon *Pten* deletion. Scale bar, 50 µm.

(Fig. 9b). The p-Akt+ tumour cells were predominantly comprises CK8-positive luminal cells (Fig. 9c,d).

## Discussion

Our study has identified a population of castration-resistant luminal epithelial cells in the prostate with regenerative and cancer initiating potential. These CARBs are distinct from a previously reported luminal castration-resistant Nkx3.1-expressing population (CARNs). This indicates functionally significant heterogeneity in the prostate luminal epithelial compartment and the existence of distinct luminal progenitor/stem cells in the castrated prostate. Further characterization of these cells, including their possible counterparts in the human prostate, malignancy and their relationship to other functionally defined luminal progenitors should be a high priority[31,32].

The lineage marking efficiency of the *Bmi1Cre$^{ER}$* driver in the prostate was relatively low. This might be related heterogeneity in Bmi1 promoter activity. Nevertheless, Bmi1+ and YFP+ cells showed many similarities in cell lineage composition and other characteristics. For example, the ratio of Bmi1+ luminal to basal cells was similar to that of YFP+ luminal to basal cells in the adult intact and regressed prostates (Supplementary Tables 1 and 2). Moreover, both Bmi1+ and YFP+ cells were resistant to apoptosis after castration (Fig. 2i–n), which means that both Bmi1+ and YFP+ cells observed in the regressed prostate are castration-resistant cells. Finally, using either Bmi1 immunostaining or YFP expression to mark CARBs showed them to be distinct from CARNs in either castrated *Bmi1-Cre$^{ER}$;R26R-YFP* mice or castrated *Nkx3.1Cre$^{ERT2}$;R26R-YFP* mice (Fig. 2h,i and Supplementary Fig. 3d–h). Altogether, these observations demonstrate that even though lineage marked YFP+ cells were detectable at low efficiency, this marking is most likely representative.

Several recent lineage tracing studies indicate that while both basal and luminal cells could serve as the cells of origin for prostate cancer, luminal cells are more sensitive to transformation[13,16]. Our studies reported here support this notion and further indicate that additional heterogeneity exists within the luminal layer that could impact on susceptibility to transformation and/or tumour aggressiveness. The high expression of the oncogenic protein Bmi1 in CARBs may make them more tumour prone than CARNs and/or result in CARB-initiated tumours being more aggressive. This is a testable hypothesis although since the currently available *Nkx3.1-*

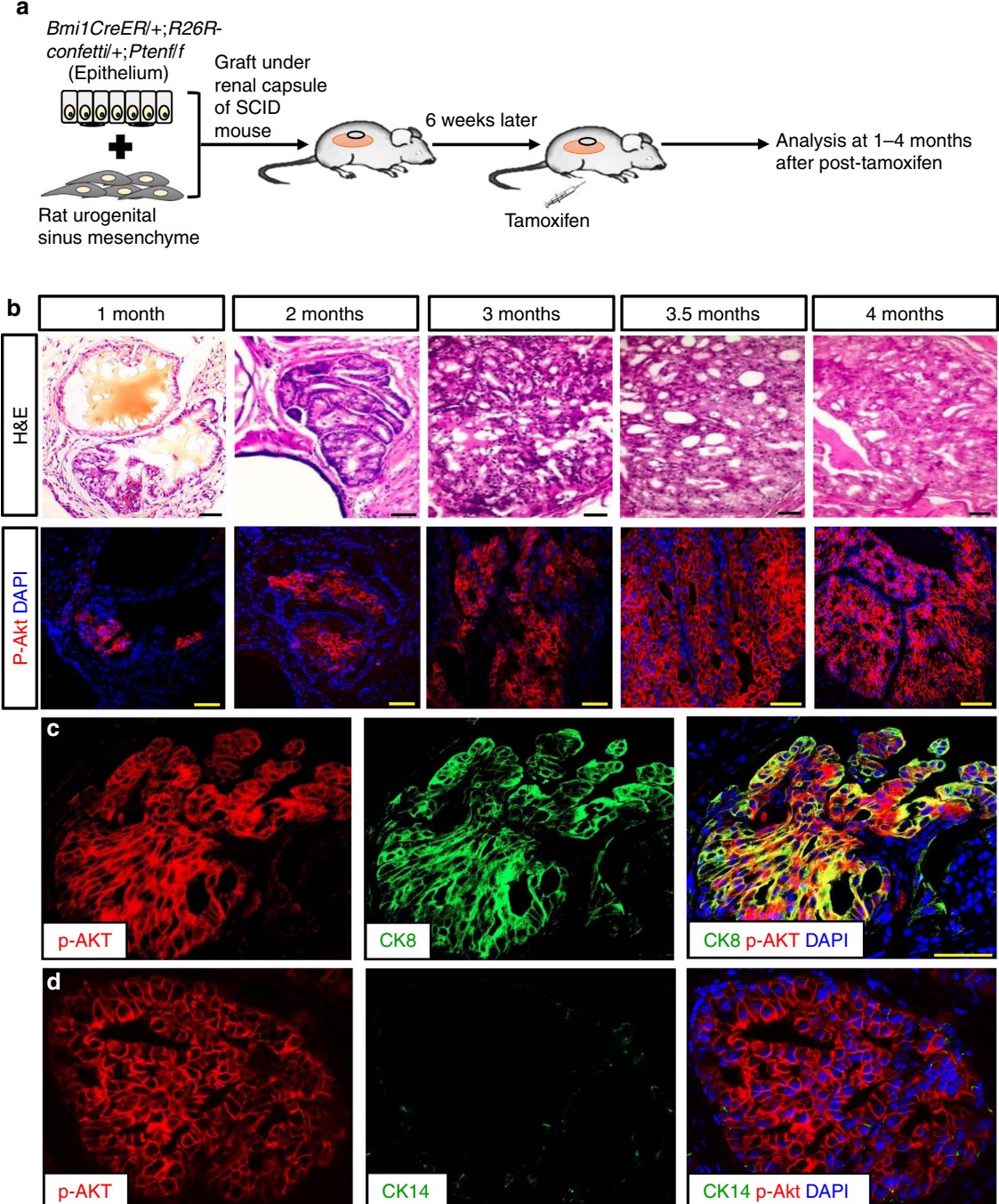

**Figure 9 | Initiation of luminal prostate cancer from Bmi1+ cells in regenerated prostate grafts.** (**a**) Scheme for regeneration of *Bmi1-Cre^{ER}*; *R26R-confetti;Ptenf/f* (*BC-Pten*) mouse prostates by tissue recombination and tumour induction. (**b**) H&E and immunoflorescence (IF) staining for p-Akt during tumour progression in tissue recombinant prostate grafts. (**c,d**) IF staining showing co-localization of CK8 (**c**) or CK14 (**d**) with p-Akt. Scale bar, 50 μm.

driven inducible Cre allele also inactivates the endogenous *Nkx3.1* gene, a proper direct test of the hypothesis will require either the generation of a new *Nkx3.1-CreER* driver which does not inactivate the endogenous *Nkx3.1* gene, or the generation of mice for analysis of CARBs where an *Nkx3.1*-null allele is introduced. The progression of *Pten*-loss initiated cancer originating in CARBs will be expected to be more rapid than in Bmi1-negative cells as Bmi1 overexpression has been shown to cooperate with *Pten* deficiency in promoting prostate tumorigenesis in transgenic mice[23].

The majority of Bmi1+ cells (as determined by Bmi1 protein expression, lineage tracing, and analysis of *Bmi1-GFP* mice) were localized to the luminal compartment of the proximal region of the prostate. A recent study has also identified a sub-population of Sca1+ cells enriched in the proximal prostate that is luminal, androgen independent and bipotential[33], suggesting that this population might have been the one identified as Bmi1+ basal in earlier studies. In any event, our study provides multiple lines of evidence showing that Bmi1 is also expressed in luminal epithelial cells of the prostate.

Prostate luminal and basal cell lineages in the adult are mainly self-sustained[13]. Our lineage tracing of Bmi1+ cells suggest that Bmi1+ luminal cells and Bmi1+ basal cells are similarly self-

sustained. Furthermore, kinetic data using BrdU incorporation showed that the percentage of newly formed YFP-labelled luminal or basal cells was about the same as that of total YFP-labelled luminal or basal cells after regeneration, suggesting that the majority of YFP-labelled luminal and basal cells give rise to the committed lineage-specified progenitors during regeneration. However, we cannot completely exclude the possibility that rare YFP-labelled basal and luminal cells that possess multipotency exist.

A major outstanding question in prostate cancer concerns the origin of CRPC. One untested notion is that cancers that arise from castration-resistant cells (for example, CARNs or CARBs) may already be 'wired' for castration resistance due to intrinsic properties of the cell of origin. Further studies including functional and molecular comparisons of CARBs and CARNs may shed light on this question. Finally, the fact that our labelling strategy could prospectively identify cells that appear to be castration resistant provides a unique tool to address the question of CRPC origins.

## Methods

**Mouse strains.** The following mouse strains were purchased from the Jackson Laboratory (Bar Harber, ME): Bmi1CreER[26], R26R-YFP[27], R26R-Confetti[34], Ptenf/f[35] and Bmi1-GFP[25]. Nkx3.1CreERT2 mice[5] were the kind gift of M. Shen (Columbia University). All mouse studies were approved by the Northwestern University Institutional Animal Care and Use Committee.

**Mouse procedures.** Adult male mice were castrated using standard techniques. Three weeks after castration, mice were administered 9 mg per 40 g tamoxifen (Sigma, St Louis, MO) suspended in corn oil by intraperitoneal injection daily for four consecutive days. Two weeks later, DHT pellets (Medisca, St-Laurent, QC, 12 mg per pellet) were implanted. Following prostate regeneration for two weeks, DHT pellets were removed to re-induce prostate regression for 2 weeks. For labelling proliferating cells, BrdU (Invitrogen, Carlsbad, CA, 100 mg kg$^{-1}$) was administered by intraperitoneal injection for 3 consecutive days or 2 h before mouse were killed.

**Tissue recombination and renal capsule grafting.** Urogenital sinus mesenchyme (UGM) cells were dissociated from embryonic day 18 (E18) embryos from pregnant Sprague Dawley rats as described previously[36,37]. Prostate glands were digested with collagenase (Invitrogen) in DMEM media with 10% FBS for 2 h at 37 °C. Subsequently, digested cells were suspended in DNase I (Sigma), and then passed through 40-μm cell strainer (Corning Inc., Corning, NY) to obtain single cells. In all, $2 \times 10^5$ dissociated cells were mixed with $1 \times 10^5$ dissociated UGM cells and re-suspended in 50 ul of 3:1 collagen:setting solution (designated as 1st tissue recombinants (TRs)). First TRs were implanted under the renal capsules of male NOD/SCID mice (4–6 weeks). Six weeks post implantation, mice were administered 9 mg per 40 g tamoxifen and grafts were collected at 1, 2, 3 or 4 months post tamoxifen treatment for analysis.

**Histology and immunofluorescence staining.** Prostates and grafts were fixed with 4% para-formaldehyde for 2 h, followed by incubation in 30% sucrose for overnight, embedded in OCT. Cryosections (4 μm) were stained with hematoxylin and eosin or specified antibodies. Primary antibodies used are listed in Supplementary Table 4. Sections were incubated in citrate buffer (pH 6) and 3% H$_2$O$_2$, followed by blocking with 10% normal goat serum (Vector Labs), incubated with primary antibodies diluted in 10% normal goat serum. Sections then were incubated with secondary antibodies labelled with Alexa Fluor 488, 594, or 647 (Invitrogen). The signals for both Bmi1 and Nkx3.1 were enhanced using tyramide amplification (PerkinElmer, Akron, OH) and horseradish peroxidase Labelled Polymer (Dako, Carpinteria, CA) system, followed by incubation with tyramide fluorescein or cyanine 3 for 6 min. Sections were counterstained with DAPI (Sigma) and mounted with ProLong Gold Antifade reagent (Invitrogen/Molecular Probes). Immunofluorescence images were visualized using fluorescent microscope or Leica A1R spectral confocal microscope.

**Statistical analysis.** Significance were evaluated using a two-tailed Student's t-test. All experiments were performed with at least three animals in each experiments.

**Data availability.** The data that support the findings of this study are available from the corresponding author on request.

Additional details about the Methods can be found in the Supplementary Methods online and details of reverse transcription–PCR primers are in Supplementary Table 5.

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

## Acknowledgements

We are grateful to M. Shen (Columbia University) for providing tissue from *Nkx3.1*^CreERT2^-*R26R-YFP* mouse and to members of the Abdulkadir Lab for valuable discussion. This work was supported by National Cancer Institute grants R01CA167966 and R01CA123484 (S.A.A.); by a Zell Family Scholar Professor award (S.A.A.); and the Grayhack Chair in Urological Research (S.A.A.).

## Author contributions

S.A.A., Y.A.Y conceived the study and wrote the manuscript.; Y.A.Y, A.N. performed the experiments and analysed the data; M.R., B.L., K.U. helped in performing the experiments and writing the manuscript; M.D. conducted evaluation of disease.

## Additional information

**Competing financial interests:** The authors declare no competing financial interests.

