## [Peer Review File · Nature Communications]

Reviewers' comments:

Reviewer #1 (Remarks to the Author):

These authors have used a lineage-tracing approach to revisit the role of Bmi1-expressing epithelial cells in prostate regeneration and as a cell of origin for prostate cancer. Interestingly, this report finds that the Bmi1-expressing cells are primarily found in the luminal compartment, differing from the conclusions by the Witte group, showing that Bmi1-expressing cells are predominantly basal (Lukacs et al. (2010)). Furthermore, the present authors show that these luminal Bmi1-expressing cells can serve as a cell of origin for cancer following deletion of Pten.

Overall, this is an interesting manuscript, but there are significant issues of methodology and interpretation that need to be addressed, as follows:

1. The authors report that Bmi1-expressing cells are mostly localized to the proximal region of the prostate, but do not explicitly define "proximal". In particular, the authors should show low-magnification immunostaining of hormonally-intact and regressed prostate sections to show the distribution of Bmi1-expressing cells along the proximal-distal axis.
2. The lineage-marking efficiency for the Bmi1-CreER driver appears to be extremely low in hormonally-intact mice. In wild-type mice, the authors report that 60% and 21.6% of luminal and basal cells express Bmi1, respectively, yet only 0.3-0.4% of epithelial cells are marked by YFP after tamoxifen induction of the BY mice. How do the authors interpret this highly inefficient marking? Is this marking representative? In contrast, the efficiency of lineage-marking appears to be much higher in the regressed prostate. What accounts for this large difference in marking efficiency between the experiments?
3. Although Figures 1 and 2 show images from each prostate lobe, it is unclear which lobe(s) is being quantitated in Fig. 1f and Fig. 2i.
4. The conclusion that Bmi1-expressing cells are quiescent is based upon the lack of Ki67 expression by lineage-marked cells. Since the prostate epithelium normally displays a low proliferation rate, is the observation of 0/199 marked cells being Ki67-positive significantly different from non-Bmi1-expressing cells? Also, in Figure 1o, why does the Ki67 immunostaining appear to be cytoplasmic?
5. The authors conclude that the Bmi1-expressing cells in the hormonally-intact prostate are castration-resistant based on their percentage increase from 0.3% to 0.64% after castration. However, this quantitation lacks a statistical analysis to determine whether 0.64% is indeed different from 0.3%. Furthermore, the authors should directly assess whether lineage-marked cells undergo apoptosis during regression.
6. The conclusion that tumors predominantly arise in the proximal prostate should be supported by low-power H&E images showing the proximal-distal distribution of neoplastic regions.
7. The authors interpret the increased size of marked cell clusters in the proximal region versus the distal prostate as indicating that proximal Bmi1-expressing cells are more efficient at cancer initiation. Can the authors exclude the possibility that clones arising in the proximal region remain coherent, whereas clones arising more distally tend to disperse, thereby giving the appearance of single marked cells? The authors could perhaps address this issue in a different way by performing grafts using proximal versus distal regions.
8. Supp. Fig. 2 summarizes data from flow-sorting experiments, but no FACS plots are presented to show primary data, and the methods used are not described.

Reviewer #2 (Remarks to the Author):

Bmi-1 is a component in the PRC-1 chromatin silencing complex that plays a critical role in regulating the activities of normal and cancer stem cells. A previous study, published in *Cell Stem Cell* (Lukacs et al., 2010) and using tissue recombination assays, has shown that Bmi-1 is important for murine prostate stem/progenitor cell activity and prostate regeneration upon experimental castration. What's unclear is whether some endogenous Bmi-1 expressing cells might be unique and be able to function as a cell of origin for prostate cancer. This study developed a lineage-tracing model to address this critical outstanding question in the field and is thus significant. Also noteworthy is that most experiments were meticulously quantified.

Major points:

1. Supplementary Fig. 1: Most Bmi1+ cells are luminal, which is inconsistent with a previous report by Lukacs et al (*Cell Stem Cell*, 2010) that suggested that the majority of Bmi-1+ cells coexpressed CK5. Unfortunately, their results with the use of Bmi1-GFP KI prostates (Supplementary Figure 2) did not provide the definitive support to their immunostaining results, as the GFP+ cells seemed to be much lower than IF staining. Perhaps use of different antibodies and appropriate negative controls might clarify these inconsistent reports.
2. The lineage-marking efficiency is extremely low with only 0.3-0.4% total epithelial cells labeled with YFP, compared to ~60% luminal (CK8+) and ~22% basal (CK5+) cells expressing endogenous Bmi1 protein in the AP. Why so low efficiency of labeling from the endogenous Bmi-1 locus? What's the difference (e.g., cell-cycle profile, clonogenic properties, regenerative capacities, etc) between the minor population of traced Bmi+ cells vs. the majority of non-traced Bmi-1+ cells?
3. It is interesting that upon castration, the % of Bmi1+ cells, or CARBs, increased to ~0.7% of the total epithelial cells. Since Bmi1+ cells seem to be largely quiescent in intact prostate, where did this doubling of the Bmi1+ cells come from? From initially weakly/negatively marked YFP+ cells or from the duplication of YFP+ cells?
4. The results about the cell cycle entry and exit (Figure 5) are potentially interesting, but the overall data is relatively weak considering the low GFP+% and subtle differences in quantification data (Fig. 5d, 5e).
5. After castration the majority (>95%) of Bmi1+ cells are luminal, but the overall Bmi1-GFP+ cells are few (0.7%). The authors assumed that the Cre-mediated PTEN deletion primarily occurred in luminal cells, but at least theoretically Cre could also be activated in basal cells, which could lead to the same observations.
6. Supplementary Fig. 1i: It's particularly interesting that CARBs are distinct from CARNs. Since NKX3.1 is exclusively expressed in luminal cells, and the Bmi-1 are also mainly localized in the luminal layer, are they co-localized before castration? If yes, why do some cells express NKX3.1 while others express Bmi1 in regressed prostate?
7. What is the difference in tumor growth kinetics of Bmi1-PTEN tumor versus published NKX3.1-PTEN tumor?

Minor points:

1. Supplementary Fig. 2d, 2e: Data presented and interpreted are confusing and may be inconsistent. Authors interpreted S2d as "GFP in both luminal and basal cell fractions". Based on the data, about 42% GFP+ cells are luminal (Sca1-CD49f-lo), and >30% GFP+ cells are basal (Sca1+CD49f-hi). However, in the S2e, authors stated that "94% of all GFP+ (i.e. Bmi1+) prostate epithelial cells were luminal (Supplementary Fig. 2e)." Are 94% GFP+ cells in luminal fraction or 94% of total GFP+ cells are luminal?
2. Their finding that CARBs are distinct from CARNs in regressed mouse prostates is interesting, which raises the possibility that many subpopulations of cells in both the luminal and basal cells

can survive castration. The main difference is that CARNs, also ranging from 0.3-0.7%, are purely luminal whereas CARBs are predominantly luminal but may also be basal. It will be very interesting to compare CARBs and CARNs in future studies.

3. Some statements are inaccurate. For example, on page 7 (first paragraph), the authors wrote that "Lesions were distinctly labeled with one of the 4 confetti colors....." but in Supplementary Fig 4c-d only GFP was shown.

4. The quality of some of the IF images could be improved (e.g., Fig. 1b-c, Fig. 3b-d).

5. There are many grammatical errors. For example, page 2, second line from the bottom, '.....mechanisms that are both dependent and independent of repression.....' should be '.....are both dependent on and independent of.....'. Another example is Supplementary Figure 3 title, which should be 'Bmi1 marks a cell population that is distinct from.....'.

Reviewer #1

Concern 1. The authors report that Bmi1-expressing cells are mostly localized to the proximal region of the prostate, but do not explicitly define "proximal". In particular, the authors should show low-magnification immunostaining of hormonally-intact and regressed prostate sections to show the distribution of Bmi1-expressing cells along the proximal-distal axis.

Response 1: We divided the prostate gland into proximal, intermediate and distal thirds. This has been clarified in the text. Low-magnification images for the distribution of Bmi1 or YFP of hormonally-intact and regressed prostate sections are also now included in the new Fig. 1d, Supplementary Fig. 1b, and Supplementary Fig. 3a-c to clearly show this.

Concern 2. The lineage-marking efficiency for the Bmi1-CreER driver appears to be extremely low in hormonally-intact mice. In wild-type mice, the authors report that 60% and 21.6% of luminal and basal cells express Bmi1, respectively, yet only 0.3-0.4% of epithelial cells are marked by YFP after tamoxifen induction of the BY mice. How do the authors interpret this highly inefficient marking? Is this marking representative? In contrast, the efficiency of lineage-marking appears to be much higher in the regressed prostate. What accounts for this large difference in marking efficiency between the experiments?

Response 2: We do make note of the low marking efficiency of the Bmi1-CreER driver in the prostate. Our first concern was with Bmi1 antibody specificity, which led us to evaluate the percentage of Bmi1+ cells using two different antibodies against Bmi1 (Supplementary Table 4) obtaining similar results. We also evaluated tissues with known Bmi1 expression patterns (duodenum, pancreas) as additional controls. The low recombination efficiency in *Bmi1-Cre^{ER}* transgenic mice might be due to Bmi1 promoter activity. Nevertheless, Bmi1+ and YFP+ cells showed many similarities in cell lineage composition and other characteristics. For example, the ratio of Bmi1+ luminal to basal cells was similar to that of YFP+ luminal to basal cells in the adult intact and regressed prostates (Supplementary Table 1 and 2). Moreover, we present new data showing that both Bmi1+ and YFP+ cells are more resistant to apoptosis after castration (Fig. 2i-n). Finally, using either Bmi1 immunostaining or YFP expression to mark CARBs showed them to be distinct from CARNs in either castrated *Bmi1-Cre^{ER};R26R-YFP* mice or castrated *Nkx3.1Cre^{ERT2};R26R-YFP* mice (Fig. 2h, i and Supplementary Fig. 3d-h). Altogether, these observations suggest that although lineage marked YFP+ cells were detectable at low efficiency, this marking is most likely representative (page9, line4).

Concern 3. Although Figures 1 and 2 show images from each prostate lobe, it is unclear which lobe(s) is being quantitated in Fig. 1f and Fig. 2i.

Response 3: Figure legends regarding Fig. 1f and Fig. 2i have been revised to clarify this.

Concern 4. The conclusion that Bmi1-expressing cells are quiescent is based upon the lack of Ki67 expression by lineage-marked cells. Since the prostate epithelium normally displays a low proliferation rate, is the observation of 0/199 marked cells being Ki67-positive significantly different from non-Bmi1-expressing cells? Also, in Figure 1o, why does the Ki67 immunostaining appear to be cytoplasmic?

Response 4: Statistical analysis showed that this difference is significant ($p < 0.01$ data from 3 mice). A clearer image for Ki67 immunostaining in Figure 1o is now shown.

Concern 5. The authors conclude that the Bmi1-expressing cells in the hormonally-intact prostate are castration-resistant based on their percentage increase from 0.3% to 0.64% after castration. However, this quantitation lacks a statistical analysis to determine whether 0.64% is indeed different from 0.3%. Furthermore, the authors should directly assess whether lineage-marked cells undergo apoptosis during regression.

Response 5: To directly examine whether YFP+ cells identified in intact mice by treating *BY* mice with tamoxifen are castration resistant, we first treated *BY* mice by tamoxifen to label the cells then castrated the mice (Fig. 3j). After castration, the fraction of YFP+ cells increased from 0.3% to 0.64% (n=253 YFP+ cells counted from a total of 39281 cells from 4 mice; p<0.001; Fig. 3k). These results suggest that tamoxifen treatment of *BY* mice before castration labels castration-resistant cells. Since we observed an increased fraction of YFP+ cells after castration compared to that in intact mice, we further investigated whether YFP+ cells are resistant to apoptosis after castration. As it had been reported that the peak of epithelial apoptosis in the mouse prostate occurred by 3-4 days after castration, the fraction of apoptotic cells were evaluated at 3 days after castration of *BY* mice following tamoxifen treatment using cleaved caspase-3 immunodetection (Fig. 3j). Interestingly, no YFP+ or Bmi1+ cells co-expressing cleaved caspase-3 were observed (Fig. 3l-n; P<0.001, n= 242 YFP+ and 862 Bmi+ cells counted from 4 mice), demonstrating the relative castration resistance of YFP+ cells prospectively marked in intact adult animals.

Concern 6. The conclusion that tumors predominantly arise in the proximal prostate should be supported by low-power H&E images showing the proximal-distal distribution of neoplastic regions.

Response 6: Low-magnification H&E images showing the proximal-distal distribution of neoplastic lesions is now included in Figure 6c.

Concern 7. The authors interpret the increased size of marked cell clusters in the proximal region versus the distal prostate as indicating that proximal Bmi1-expressing cells are more efficient at cancer initiation. Can the authors exclude the possibility that clones arising in the proximal region remain coherent, whereas clones arising more distally tend to disperse, thereby giving the appearance of single marked cells? The authors could perhaps address this issue in a different way by performing grafts using proximal versus distal regions.

Response 7: We only observed singly labelled confetti cells in both distal and proximal prostates of *Bmi1-Cre^{ER};R26R-confetti* intact mice early after tamoxifen induction, making it unlikely that the differences in clone size observed are due to existing coherent clones in the proximal prostate. These data are now shown in Supplementary Fig. 4c, d.

Concern 8. Supp. Fig. 2 summarizes data from flow-sorting experiments, but no FACS plots are presented to show primary data, and the methods used are not described.

Response 8: FACS plots and methods are now included in Supplementary Fig. 2b-d. We have also added new RT-PCR data from sorted basal and luminal cells showing that CD49f+Sca1- luminal cells robustly express Bmi1.

Reviewer #2

Concern 1. Supplementary Fig. 1: Most Bmi1+ cells are luminal, which is inconsistent with a previous report by Lukacs et al (Cell Stem Cell, 2010) that suggested that the majority of Bmi1+ cells coexpressed CK5. Unfortunately, their results with the use of Bmi1-GFP KI prostates (Supplementary Figure 2) did not provide the definitive support to their immunostaining results, as the GFP+ cells seemed to be much lower than IF staining. Perhaps use of different antibodies and appropriate negative controls might clarify these inconsistent reports.

Response 1: We were indeed surprised to find that most Bmi1+ cells are luminal. We therefore used two distinct Bmi1 antibodies in validated immunohistochemical assays, RT-PCR analysis of Bmi1 expression in FACS-sorted prostate epithelial luminal and basal cells, a *Bmi1-GFP* knockin mouse and lineage tracing with a *Bmi1CreER* driver to show that Bmi1 is expressed in luminal cells. Furthermore, our functional data on luminal CARBs indicate that Bmi1 marks progenitor luminal cells that can serve as a prostate cancer cell of origin. We do note that although the report by Lukacs et al emphasized the functional role of Bmi1 in basal cells, their study did not rule out a role for Bmi1

expression in luminal cells. Conversely, our study did not exclude a functional role for Bmi1 in prostate basal cells.

Concern 2. The lineage-marking efficiency is extremely low with only 0.3-0.4% total epithelial cells labeled with YFP, compared to ~60% luminal (CK8+) and ~22% basal (CK5+) cells expressing endogenous Bmi1 protein in the AP. Why so low efficiency of labeling from the endogenous Bmi1 locus? What's the difference (e.g., cell-cycle profile, clonogenic properties, regenerative capacities, etc) between the minor population of traced Bmi+ cells vs. the majority of non-traced Bmi-1+ cells?

Response 2: We do make note of the low marking efficiency of the Bmi1-CreER driver in the prostate. Our first concern was with Bmi1 antibody specificity, which led us to evaluate the percentage of Bmi1+ cells using two different antibodies against Bmi1 (Supplementary Table 4) obtaining similar results. We also evaluated control tissues with known Bmi1 expression patterns (duodenum, pancreas) as additional controls. The low recombination efficiency in *Bmi1-Cre^{ER}* transgenic mice might be due to Bmi1 promoter activity. Nevertheless, Bmi1+ and YFP+ cells showed many similarities in cell lineage composition and other characteristics. For example, the ratio of Bmi1+ luminal to basal cells was similar to that of YFP+ luminal to basal cells in the adult intact and regressed prostates (Supplementary Table 1 and 2). Moreover, we present new data showing that both Bmi1+ and YFP+ cells were resistant to apoptosis after castration (Fig. 2i-n), which means that both Bmi1+ and YFP+ cells observed in the regressed prostate are castration-resistant cells. Finally, using either Bmi1 immunostaining or YFP expression to mark CARBs showed them to be distinct from CARNs in either castrated *Bmi1-Cre^{ER};R26R-YFP* mice or castrated *Nkx3.1Cre^{ERT2};R26R-YFP* mice (Fig. 2h, i and Supplementary Fig. 3d-h). Altogether, these observations demonstrate that even though lineage marked YFP+ cells were detectable at low efficiency, this marking is most likely representative (page9, line4).

Concern 3. It is interesting that upon castration, the % of Bmi1+ cells, or CARBs, increased to ~0.7% of the total epithelial cells. Since Bmi1+ cells seem to be largely quiescent in intact prostate, where did this doubling of the Bmi1+ cells come from? From initially weakly/negatively marked YFP+ cells or from the duplication of YFP+ cells?

Response 3: We believe this observation is due to the higher relative resistance of CARBs to castration. To directly examine whether YFP+ cells identified in intact mice by treating *BY* mice with tamoxifen are castration resistant, we first treated *BY* mice by tamoxifen to label the cells then castrated the mice (Fig. 3j). After castration, the fraction of YFP+ cells increased from 0.3% to 0.64% (n=253 YFP+ cells counted from a total of 39281 cells from 4 mice; p<0.001; Fig. 3k). These results suggest that tamoxifen treatment of *BY* mice before castration labels castration-resistant cells. Since we observed an increased fraction of YFP+ cells after castration compared to that in intact mice, we further investigated whether YFP+ cells are resistant to apoptosis after castration. As it had been reported that the peak of epithelial apoptosis in the mouse prostate occurred by 3-4 days after castration, the fraction of apoptotic cells were evaluated at 3 days after castration of *BY* mice following tamoxifen treatment using cleaved caspase-3 immunodetection (Fig. 3j). Interestingly, no YFP+ or Bmi1+ cells co-expressing cleaved caspase-3 were observed (Fig. 3l-n; P<0.001, n= 242 YFP+ and 862 Bmi+ cells counted from 4 mice), demonstrating the relative castration resistance of YFP+ cells prospectively marked in intact adult animals.

Concern 4. The results about the cell cycle entry and exit (Figure 5) are potentially interesting, but the overall data is relatively weak considering the low GFP+% and subtle differences in quantification data (Fig. 5d, 5e).

Response 4: We have edited this section to avoid over-interpretation of the data.

Concern 5. After castration the majority (>95%) of Bmi1+ cells are luminal, but the overall Bmi1-GFP+ cells are few (0.7%). The authors assumed that the Cre-mediated PTEN deletion primarily occurred in

luminal cells, but at least theoretically Cre could also be activated in basal cells, which could lead to the same observations.

Response 5: Although *Pten* was deleted in some *Bmi1*-expressing basal cells, the resulting p-Akt+ basal cells were detected specifically only in normal looking glandular structures (Supplementary Fig. 4h), suggesting *Pten* deletion in *Bmi1*+ basal cells does not efficiently initiate transformation. Nonetheless, we agree with the reviewer that alternative possibilities such as rapid luminal differentiation of *Pten*-deleted basal cells fated to develop PIN/cancer cannot be completely ruled out and it is best not to be dogmatic. We have made note of this caveat in our discussion of the data.

Concern 6. Supplementary Fig. 1i: It's particularly interesting that CARBs are distinct from CARNs. Since NKX3.1 is exclusively expressed in luminal cells, and the Bmi-1 are also mainly localized in the luminal layer, are they co-localized before castration? If yes, why do some cells express NKX3.1 while others express Bmi1 in regressed prostate?

Response 6: This is an interesting question which however is difficult to answer at present. This is due to the fact that *Nkx3.1* is expressed in virtually all luminal cells in intact adult mouse prostate, and there is no way at present to determine which of these cells are CARNs prior to castration. Thus before castration, *Nkx3.1* and *Bmi1* maybe co-localized in cells but there is no evidence these cells are stem/progenitor cells. After castration however, *Nkx3.1* and *Bmi1* are expressed in distinct luminal cells that are functionally prostate progenitor cells susceptible to tumor initiation.

Concern 7. What is the difference in tumor growth kinetics of Bmi1-PTEN tumor versus published NKX3.1-PTEN tumor?

Response 7: The *NKX3.1-PTEN* tumors are also deficient in *Nkx3.1* tumor suppressor protein since the *Nkx3.1-CreER* allele inactivates the endogenous *Nkx3.1* gene. By contrast, the *Bmi1-CreER* driver was made by insertion into the 3'UTR of the mouse *Bmi1* gene and does not affect expression of the endogenous *Bmi1* gene. Thus a comparison of the *Bmi1-PTEN* and *NKX3.1-PTEN* tumor growth kinetics is not straightforward. We are currently generating and analyzing mice with equivalent genetic changes (i.e. by introducing *Nkx3.1* heterozygosity into the *Bmi1-PTEN* mice) to address this question. Similarly, we are generating similar mice to compare the molecular profiles of non-transformed CARBs and CARNs (i.e. *Bmi1CreER;Nkx3.1+/-;R-YFP* vs *NkxCreER;R-YFP* mice).

Minor points:

1. Supplementary Fig. 2d, 2e: Data presented and interpreted are confusing and may be inconsistent. Authors interpreted S2d as "GFP in both luminal and basal cell fractions". Based on the data, about 42% GFP+ cells are luminal (Sca1-CD49f-lo), and >30% GFP+ cells are basal (Sca1+CD49f-hi). However, in the S2e, authors stated that "94% of all GFP+ (i.e. Bmi1+) prostate epithelial cells were luminal (Supplementary Fig. 2e)." Are 94% GFP+ cells in luminal fraction or 94% of total GFP+ cells are luminal?

Response: We have now clarified this in the description and presentation of experiments in Supplementary Fig. 2, clearly indicating what fraction of luminal or basal cells are GFP+.

2. Their finding that CARBs are distinct from CARNs in regressed mouse prostates is interesting, which raises the possibility that many subpopulations of cells in both the luminal and basal cells can survive castration. The main difference is that CARNs, also ranging from 0.3-0.7%, are purely luminal whereas CARBs are predominantly luminal but may also be basal. It will be very interesting to compare CARBs and CARNs in future studies.

Response: We agree. We are generating mice to compare the molecular profiles of non-transformed CARBs and CARNs (i.e. *Bmi1CreER;Nkx3.1+/-;R-YFP* vs *NkxCreER;R-YFP* mice).

3. *Some statements are inaccurate. For example, on page 7 (first paragraph), the authors wrote that "Lesions were distinctly labeled with one of the 4 confetti colors....." but in Supplementary Fig 4c-d only GFP was shown.*

Response: This has been corrected.

4. *The quality of some of the IF images could be improved (e.g., Fig. 1b-c, Fig. 3b-d).*

Response: Images for Fig. 1b-c and Fig. 3b-d have been changed.

5. *There are many grammatical errors. For example, page 2, second line from the bottom, '.....mechanisms that are both dependent and independent of repression.....' should be '.....are both dependent on and independent of.....'. Another example is Supplementary Figure 3 title, which should be 'Bmi1 marks a cell population that is distinct from.....'.*

Response: Grammatical errors have been corrected.

REVIEWERS' COMMENTS:

Reviewer #1 (Remarks to the Author):

This revised manuscript has been revised significantly to address previous critiques. Most of the issues have been addressed satisfactorily, but there are several remaining minor points that should be resolved prior to publication.

1. Previously, the question was raised as to why the lineage-marking efficiency of the Bmi1-CreER driver was much higher in the regressed prostate, but the authors did not directly address this issue in their rebuttal. In new data added for this revision, the authors show that YFP-positive cells marked in the intact prostate are more resistant to apoptosis after castration, but it remains unclear whether they believe that this difference provides a full explanation of the higher marking efficiency.
2. The authors show that YFP-positive and Bmi1-positive cells do not co-express cleaved caspase 3 at 3 days after castration, but do not provide a control quantitation for the percentage of apoptotic cells in the overall population.
3. The revised Figure 6 now includes a low-power image to support the conclusion that tumors predominantly arise in the proximal region. However, given that the differences shown are not extremely obvious, and that tumors also arise distally, the authors should provide some quantitation for the percentage of tumors arising proximally versus distally.

Reviewer #2 was unable to assess the revised manuscript. Reviewer#1 comments on the authors' response to Reviewer #2:

In re-reading this manuscript and the corresponding reviews, I believe that the authors have adequately addressed the previous concerns raised by Reviewer 2. My only additional comment is as follows:

Both reviewers raised the issue of extremely low marking efficiency of the Bmi1CreER driver, and raised the question of whether the YFP-positive cells are completely representative of the entire population of Bmi1-positive cells. The authors address this issue in the new paragraph on p. 9, but given the lines of evidence presented, perhaps the conclusion that the "marking is most likely representative" is overstated.

Reviewer #1

Concern 1. Previously, the question was raised as to why the lineage-marking efficiency of the Bmi1-CreER driver was much higher in the regressed prostate, but the authors did not directly address this issue in their rebuttal. In new data added for this revision, the authors show that YFP-positive cells marked in the intact prostate are more resistant to apoptosis after castration, but it remains unclear whether they believe that this difference provides a full explanation of the higher marking efficiency.

Response 1: We observed a 2-fold difference in marking efficiency in intact versus castrated mice with the Bmi1-CreER driver. We believe this could be explained by the fact that CARBs are resistant to castration, which results in them being “enriched” in the castrated prostate.

Concern 2. The authors show that YFP-positive and Bmi1-positive cells do not co-express cleaved caspase 3 at 3 days after castration, but do not provide a control quantitation for the percentage of apoptotic cells in the overall population.

Response 2: We have now added the overall percentage of apoptotic cells by 3 days of castration in the Results section (page 5): 4.8%, n=387 c-casp3+ cells counted from a total of 7994 cells from 4 mice.

Concern 3. The revised Figure 6 now includes a low-power image to support the conclusion that tumors predominantly arise in the proximal region. However, given that the differences shown are not extremely obvious, and that tumors also arise distally, the authors should provide some quantitation for the percentage of tumors arising proximally versus distally.

Response 3: To be clear, at 2 months after Pten deletion in CARBs, we only observed p-Akt+ lesions in the proximal region but not in the distal region. The differences were stark (Fig. 6c-r). At 4 months of age, lesions were observed in both the proximal and distal regions but the proximal region has more advanced and larger lesions. Our confetti labeling data in Bmi1-CreER;R-Confetti;Pten^{flox/flox} mice showed this phenomenon quantitatively (Fig. 7). Nonetheless we have now added more data quantitating p-Akt+ lesions in the proximal and distal prostates at 2 months and 4 months (page 7). The results further confirm that that proximal lesions are larger and more frequent at both 2 months and 4 months.

Sincerely,

Sarki Abdulkadir